# PRECIS-2 used as an implementation science tool for global environmental health: A cross-sectional evaluation of the Ecolectivos study protocol to reduce burning of household plastic waste in rural Guatemala

**Lisa M. Thompson** [1,2‡] *, **Annalyse Ferguson** [1‡], **Hina Raheel** [1], **Amy E. Lovvorn** [2], **Mayari Hengstermann-Artiga** [3], **Maria Renee Lopez** [3], **Melinda Higgins** [1], **Eri Saikawa** [2,4], **Margaret A. Handley** [5,6]

1 Nell Hodgson Woodruff School of Nursing, Emory University, Atlanta, GA, United States of America, 2 Gangarosa Department of Environmental Health, Emory University Rollins School of Public Health, Atlanta, GA, United States of America, 3 Center for Health Studies, Universidad del Valle de Guatemala, Guatemala City, Guatemala, 4 Department of Environmental Sciences, Emory University, Atlanta, GA, United States of America, 5 Department of Epidemiology and Biostatistics, University of California San Francisco, San Francisco, CA, United States of America, 6 PRISE Center (Partnerships for Research in Implementation Science for Equity), University of California San Francisco, San Francisco, CA, United States of America

‡ LMT and AF are co-first authors on this work.
* Lisa.thompson@emory.edu

## Abstract

### Background

Randomized controlled trials (RCTs) that evaluate the efficacy of an intervention remain underutilized in community-based environmental health research. RCTs that use a pragmatic design emphasize the effectiveness of interventions in complex, real world settings. Pragmatic trials may be especially relevant when community-based interventions address social and environmental determinants that threaten health equity. The revised Pragmatic Explanatory Continuum Indicator Summary (PRECIS-2) is a validated tool developed in 2015 by trialists to ensure that clinical trials are designed to fit their intended purpose, with an assessment of applicability of the trial results to specific contexts. The purpose of this cross-sectional study was to ask Ecolectivos study investigators and external implementation scientists to evaluate the Ecolectivos study protocol using the PRECIS-2 tool prior to the launch of the trial. Ecolectivos is an implementation science study, using a village-level cluster randomized controlled trial design, to assess a behavioral intervention to reduce household plastic waste burning in rural Guatemala.

### Methods

We invited 60 researchers to participate in an online survey between February 2022 and January 2023. Respondents were asked to review the Ecolectivos study protocol and provide scores for the nine PRECIS-2 domains (eligibility, recruitment, setting, organization,

**Data Availability Statement:** All files are published and freely available in a public repository (Dataverse) through Emory library's Dataverse at https://doi.org/10.15139/S3/WF83RR

**Funding:** The Ecolectivos Trial is supported by the National Institute of Environmental Health Sciences (NIEHS) of the National Institutes of Health under Award Number R01ES032009. The content is solely the responsibility of the authors and does not necessarily represent the official views of the National Institutes of Health. The funders had no role in study design, data collection and analysis, decision to publish, or preparation of the manuscript.

**Competing interests:** The authors declare that they have no competing interests.

**Abbreviations:** IQR, Interquartile Range; PRECIS-2, Pragmatic Explanatory Continuum Indicator Summary; RCT, Randomized controlled trial.

flexibility-*delivery*, flexibility-*adherence*, follow-up, primary outcome, and primary analysis), with short responses explaining their score. The PRECIS-2 tool is used to assess the degree of pragmatism, ranked on a five-point Likert scale from very explanatory (Checkley W, 2022) to very pragmatic (Ashcraft LE, 2024). Descriptive statistics were used to compare responses between Ecolectivos investigators and external evaluators.

## Results

Twenty-five respondents provided data. Among the nine domains, four were rated as pragmatic—eligibility, setting, flexibility-*delivery*, and primary analysis. Four were evaluated to be equally pragmatic as explanatory—recruitment, organization, flexibility-*adherence*, and primary outcome. One domain was primarily explanatory in nature—follow-up. Only one domain, eligibility, was statistically significantly different between Ecolectivos investigators and external evaluators, demonstrating that the two groups were broadly consistent in their opinions in eight of the nine study domains. Using the PRECIS-2 tool, we found that our study protocol was viewed as more pragmatic than explanatory, providing evidence to support the pragmatic approach of the Ecolectivos study goals, which is to reduce burning of plastic waste and plastic use in community settings using a behavioral intervention.

## Conclusions

By evaluating the degree of pragmatism within the nine domains, PRECIS-2 guides investigators to think about the applicability of potential results. Investigator assessment and communication regarding intervention protocols for community-level environmental interventions, their degree of pragmatism, and external validity are important for identifying strategies to address complex community problems. Our findings contribute to the growing body of literature that addresses greater research utility through pragmatic trial design, tying community environmental health interventions to the rigor of implementation science strategies.

## Background

Testing the efficacy and the effectiveness of interventions using randomized controlled trial (RCT) designs are the "bread and butter" of clinical research. However, RCT designs remain underutilized within environmental health research, with observational study designs most frequently used [1]. RCTs should play a more important role in advancing environmental health research by identifying more efficacious interventions to reduce environmental exposures and mitigate health risks [2]. A pragmatically designed RCT conducted in community settings is one approach that can generate externally valid results for complex interventions in resource-constrained settings [3]. This is especially relevant in community-based environmental health research, where implementation researchers must contend with compounding social (e.g., inclusion/exclusion; racism; access to education and healthcare), environmental (e.g., neighborhood, housing; proximity to pollution), and economic (e.g., employment, income) determinants that threaten health equity [4] and undermine environmental justice [5]. A pragmatic design is described as occurring under "usual" conditions; in comparison, an explanatory design aligns with a more rigidly controlled trial under ideal conditions. If a pragmatic

RCT design measures effectiveness, then an explanatory RCT design measures efficacy [6]. In practice, pragmatic and explanatory exist on a continuum and do not represent a rigid dichotomy [7]. Since maximizing internal validity is a key aspect of the classic RCT design, researchers from diverse disciplines have started to stress the importance of designing pragmatic trials, with the goal of greater external validity [7, 8]. Such trials may include standardization but should maintain enough structured flexibility to adapt to local contexts and unique participant needs that intervention fidelity is still achieved [3]. Although it is impossible for RCTs to be applicable or relevant for all populations and settings, pragmatic RCTs have the benefit of aiding researchers and practitioners in recognizing which interventions deserve priority attention for their specific settings based on transparent user applicability.

Loudon et al.'s (2015) revised Pragmatic Explanatory Continuum Indicator Summary (PRECIS-2) is a validated implementation tool developed to ensure that trials are designed to fit their intended purpose, with the degree of pragmatism ranked on a five-point Likert scale from very explanatory (1) to very pragmatic (5) [7]. The nine domains of PRECIS-2 include: eligibility, recruitment, setting, organization, flexibility in delivery, flexibility in adherence, follow-up, primary outcome, and analysis. By guiding investigator evaluation of the degree of pragmatism within these nine domains of trial design and conduct, PRECIS-2 aids the assessment of the consequences of study design and the utility/applicability of the potential results for the intended audience. This requires trialists to develop a foundational agreement on the design and purpose of the trial, prior to deciding how pragmatic the intervention domains may be [9]. While this tool is optimally used prospectively by study investigators, it has also been used retrospectively in systematic reviews [10].

The rationale for the Ecolectivos study stems from a global environmental problem, the ubiquitous presence of plastic waste in low- and middle-income countries that lack municipal waste management [6]. Household burning of plastic waste in indoor cooking fires and outdoor trash fires is a commonly cited response to this predicament. Since plastics burn quickly and release intense heat, plastic waste is also used to kindle cooking fires within household hearths [11]. While many studies on household air pollution from cooking, lighting, and heating sources have been conducted [12–14], personal exposures to burning plastic waste in these contexts has not been well characterized [15]. Plastic combustion releases carcinogenic compounds such as dioxin, as well as bisphenols and phthalates, known to be disruptive of endocrine and reproductive function as well as fetal neurodevelopment [16]. The Ecolectivos study aims to address gaps in this area of research by assessing potential reductions in personal exposures to the burning of household plastic waste, a complex, yet poorly described, environmental health risk [15, 17].

Influencing and understanding behavior change is an important motivation when developing solutions to stop the burning of plastic waste [18–20], and solutions are more difficult to implement in low-resource settings. In rural Xinca communities in Santa Maria Xalapán in Jalapa, Guatemala, where the Ecolectivos study is being conducted, communities are experiencing plastic proliferation without the established infrastructure for its organized elimination [21]. With the priority of identifying local solutions that center on external validity and sustainable real-world application, the Ecolectivos research study aims to promote community-wide behavior change over a five-year study period through a range of community-led environmental intervention strategies to reduce household plastic waste burning and thereby decrease health risk burdens. Theory and evidence-based behavior change research are necessary to guide these interventions. A pragmatic trial design and protocol evaluation offer greater research utility by providing a thorough evaluation of the implementation science approaches. Tools like PRECIS-2 are useful in addressing the consequences of the designed study and its degree of pragmatism [7, 22].

The Ecolectivos study is an implementation research study using a village-level cluster RCT. It is a type 1 hybrid-effectiveness-implementation study that measures effectiveness as the primary study aim, with additional measures related to implementation, such as reach and acceptability, also evaluated [23, 24]. The study is theoretically guided by the related implementation science frameworks, the COM-B model and the Theoretical Domains Framework (TDF) [22, 25]. The primary goal of the Ecolectivos study is to evaluate the uptake and sustainability of intervention strategies to reduce, recycle, and repurpose plastic that will lead to reductions in household-level plastic burning in communities [26]. Briefly, the objectives of the study are to: 1) implement and evaluate intervention strategies that address plastic waste burning; and 2) evaluate the effect of these behavioral interventions on urinary biomarkers of exposure to plastic waste burning (bisphenols, phthalates, polycyclic aromatic hydrocarbons) and personal airborne fine particulate matter ($PM_{2.5}$) and black carbon (BC). There are two phases to the study: During the formative phase (year 1), we will conduct a baseline assessment to characterize plastic waste disposal practices and identify villages and women of reproductive age for the main trial phase. During the main trial phase (years 2–4), we will enroll 400 women of reproductive age from eight intervention and eight control villages into a biomonitoring study. Our team will conduct 12-week educational workshops in the eight intervention villages. Women in the biomonitoring group in intervention villages, as well as other community members, will be invited to attend the workshops. The first eight weeks consist of standard educational workshops. The last four weeks are used to define village-specific environmental intervention strategies that will be directed by a subset of women participating in the working groups. Based on participatory action research (e.g., community clean-ups, recycling), the purpose of these strategies is to reduce the burning of household plastic (see **Table** 1 for study overview). The groundwork for the development and content of the educational workshops was established over two years during our community-engaged pilot work in one community. Workshops were created by an anthropologist and evaluated using a theory-informed behavioral approach. Other examples that are similar to our approach looked at action plans for metal recyclers and communities living near polluting recycling facilities in Texas [27], a participatory action study that aimed to develop a sustainable waste disposal model in river communities in Thailand [28], and participant-driven littering campaigns in South African schools [29]. The formal evaluation of Ecolectivos' behavioral intervention and trial design is an essential step to ensure that the study is appropriately designed to achieve its intended purpose of reducing plastic waste burning through targeted behavioral change. The objectives of the present paper are to: (1) use the validated PRECIS-2 tool to evaluate investigator perceptions of the trial, and (2) determine the trial's degree of pragmatism and applicability. Our effort contributes to the growing body of literature tying community environmental health interventions to the rigor of implementation science, with the intention of addressing greater research utility through a more pragmatic trial design.

## Methods

### Study design and sample

We conducted a cross-sectional exploratory survey administered via targeted email to Ecolectivos investigators, as well as external evaluators familiar with implementation research who were not participating in the Ecolectivos trial (hereafter referred to as "external evaluators"). Between February 2022 and January 2023, surveys were sent out to 60 researchers across 20 academic institutions, comprised of 46 external evaluators and 14 Ecolectivos investigators. To improve the response rate, three reminders were sent during this timeframe. All survey instruments were administered in English.

**Table 1. Ecolectivos study activities.**

| | | Formative (Year 1) | |
|---|---|---|---|
| **Aim 1** | Rapid Assessment | 1,600 households 37 sectors | • household/demographic characteristics • waste management/recycling survey • ID 400 women for biomonitoring study • ID one community for working group pilot |
| | CAB | 15 members | • meet monthly (years 1–5) |
| | Working group refinement | 20–30 households that burn plastic waste 20 key informant interviews | • open-ended survey (BCW questions) • participant observation in households •stakeholders, government and non-government workers |
| | Working group pilot | 4 fieldworkers 50 members; 1 village | • train to deliver working group content • finalize curriculum/procedure manuals |
| | | **Main (Years 2–4)** | |
| **Aim 1** | Community Working groups *12-weeks (Months 1–3)* | 400 participants in 8 intervention villages | • 8 core modules & 4 periphery modules • evaluate intervention fidelity • weekly surveys on behavior adoption r/t classes • assess COM-B framework and TDF domains |
| | Intervention activities evaluation *Months 6, 9 & 12* | 24–40 participants in 8 intervention villages 3–5 women in each village in charge of implementation activities | • address bottlenecks to strengthen implementation • assess RE-AIM outcomes |
| | Focus group evaluation *Month 12* | 6–8 focus groups 60 working group participants | • working group evaluation • assess RE-AIM outcomes |
| **Aim 2** | Biomonitoring Study *Baseline, Months 4 & 12* | 200 intervention & 200 control participants in 16 villages | • household/demographic characteristics • plastic behaviors (e.g., burning, recycling) • air pollution exposures (24-hour personal $PM_{2.5}$) • health related quality of life; general self-efficacy • urinary biomarkers (bisphenols, phthalates, PAHs) |
| | Plastic waste collection *Baseline, Months 4 & 12* | 80 intervention & 80 control participants in 16 villages | • collect household plastic trash they would have burned for one week |
| | | **Evaluation and Dissemination (Year 5)** | |
| | Report-back of findings Capacity building activities | community participants CAB members local stakeholders regional/nat'l policy-makers | • recruitment & retention of participants • evaluate individual and community strategies • evaluate implementation fidelity • disseminate intervention strategies |

The survey included questions related to the PRECIS-2 domains for the evaluation of the original Ecolectivos study protocol. The original PRECIS-2 paper [7], an educational Power-point presentation, and a paper using the PRECIS-2 tool for a global air pollution trial was provided so that reviewers could understand our intended approach [1]. We asked reviewers to rank their perceived degree of pragmatism in the PRECIS-2 nine domains of trial design and conduct, using a five-point Likert scale: very explanatory (1), explanatory (2), equally explanatory/pragmatic (3), pragmatic (4), or very pragmatic (5). We also asked them to provide short answers explaining their score choices. We also asked participants two questions, "on a scale of 1–10 (1 = low, 10 = high), how knowledgeable are you about the discipline of Implementation and Dissemination Science?" and "on a scale of 1–10 (1 = low, 10 = high), how much experience have you had working with randomized controlled trials?"

The nine domains of PRECIS-2 tool described briefly are: 1) *eligibility* (trial participants similar to those who would receive the intervention if it was usual care); 2) *recruitment* (effort needed to recruit participants beyond what would be used in usual care settings); 3) *setting* (setting similar to usual care setting); 4) *organization* (resources, provider expertise, and organization of care delivery in intervention different from usual care); 5) *flexibility in delivery*

(flexibility in how the intervention is delivered versus flexibility in usual care); 6) *flexibility in adherence* (flexibility in how participants are monitored to adhere to the intervention versus usual care); 7) *follow-up* (follow-up of participants in trial similar to typical follow-up in usual care); 8) *primary outcome* (trial's primary outcome directly relevant to participants); and 9) *analysis* (all data included in the analysis of the primary outcome). The developers of the PRECIS-2 tool found the tool to have good face validity, discriminant validity (able to determine pragmatism across a broad spectrum of trial protocols) and interrater reliability (intraclass correlation coefficient > 0.65) [30].

All participants provided written informed consent to participate. They entered their responses into an anonymous, online data collection tool, which was stored on a password protected, secure cloud-based server. We did not collect personal identifying information; instead we collected gender, race, ethnicity, and age as categorical demographic characteristics. We did not collect data on educational degree (all possessed a doctorate degree (e.g., PhD or MD)) or specific institution.

## Ethics approval and consent to participate

All participants provided written informed consent to participate. Responses did not include identifying information about the respondent. The study protocol has been reviewed and approved by institutional review boards (IRBs) at the Universidad del Valle de Guatemala (245-05-2021) and Emory University (#00002412), with Reliance Agreements the University of Georgia (#00005189) and the University of California at San Francisco (21–35706). The study has been registered with ClinicalTrials.gov (Identifier NCT05130632).

## Data analysis

We summarized PRECIS-2 Likert scores using medians (interquartile ranges (IQR), representing the 25th and 75th percentile scores) for each domain. We evaluated the distributions of the data to assess whether assumptions of normality were met. Since the data were not normally distributed, we tested the differences in scores between the Ecolectivos investigators and the external evaluators using the nonparametric Mann Whitney-U test. We used Excel and SPSS version 29.0 for data analyses. Given the nature of this small study, we did not conduct a power analysis for the quantitative analysis.

The purpose of the qualitative analysis was to describe themes that drove the PRECIS-2 domain scores. The short responses supporting why each respondent chose their score for each of the nine domains were analyzed thematically and grouped to illustrate common, recurrent patterns in the qualitative data that enriched the quantitative findings [31]. Themes were identified and exemplars were then extracted to illustrate the theme and integrated into the results. Using online surveys to retrieve descriptive qualitative findings has been found to provide valid information for researchers [32].

## Results

A total of 25 investigators responded to the online survey (Table 2). Of the 46 external evaluators who were invited, 16 completed the evaluation (35% response rate). Of the 14 Ecolectivos investigators who were invited, 9 completed the evaluation (64% response rate). The demographic characteristics showed that more females (88%), and non-Latino white respondents (68%), participated in the survey.

Reponses about knowledge of Implementation and Dissemination Science and experience with randomized controlled trials did not yield statistically significant differences in responses between Ecolectivos investigators and external evaluators (Table 3).

**Table 2. Descriptive characteristics of the study participants (n = 25).**

| | | Count | Percentage |
|---|---|---|---|
| Gender | *Male* | 3 | 12% |
| | *Female* | 22 | 88% |
| Age, years | *25–34* | 6 | 24% |
| | *35–44* | 10 | 40% |
| | *45–54* | 6 | 24% |
| | *55–64* | 3 | 12% |
| Race | *Asian* | 2 | 8% |
| | *Black* | 2 | 8% |
| | *Other* | 4 | 16% |
| | *White* | 17 | 68% |
| Ethnicity | *Latino* | 4 | 16% |
| | *Non-Latino* | 21 | 84% |
| Ecolectivos team member | *No* | 16 | 64% |
| | *Yes* | 9 | 36% |

Of the nine domains on the PRECIS-2 scale, four were rated as rather pragmatic (score = 4)—*flexibility of delivery*, *setting*, *eligibility*, and *analysis* (see **Table 4**). Four of the nine domains were evaluated to be equally as pragmatic as explanatory (score = 3)—*organization*, *recruitment*, *outcome*, and *flexibility of adherence*. One domain was assessed to be primarily explanatory (score = 1)—*follow-up*. Only one domain, *eligibility*, had a statistically significant different rating between Ecolectivos investigators and external evaluators (**Table 4 and Fig 1**). The feedback from the descriptive comments provided by all evaluators provided further insight into the ratings for each domain.

## 1. Eligibility criteria: *To what extent are the participants in the trial similar to those who would receive this intervention if it was part of usual care*?

The median response for eligibility among all respondents was 4 (out of a range of 5), or rather pragmatic (IQR 3, 5). Narrative feedback highlighted that the protocol's broad inclusion criteria was the top rationale for receiving a primarily pragmatic ranking. One external reviewer who viewed the trial as "rather pragmatic" stated, *"as community level workshops have no inclusion or exclusion criteria, the intervention itself is quite pragmatic. Exclusion of pregnant women or tobacco-using. . .. for the urine and personal air pollution for the [biomonitoring women] samples (as opposed to lack of criteria for questionnaires to workshop attendees) is why I selected "rather" instead of "very" pragmatic."* Several respondents noted that the intervention would likely be reflective of actual community conditions since there are minimal exclusion criteria. However, although the eligibility criteria make the sample likely representative of the Jalapa, Guatemala population, evaluators disagreed about whether this was a limitation or benefit to overall pragmatism. While Ecolectivos members stated the trial was more explanatory (median, 2 [IQR 2,3]), external evaluators scored the trial as more pragmatic (median, 4 [IQR 4,5]) (p-value, < 0.001), showing a divergence of opinions. One Ecolectivos co-investigator stated, *"For the main trial, there are a fair number of eligibility criteria, and the willingness to attend 12 working group sessions may exclude a lot of people."*

**Table 3. Differences in knowledge of implementation science and experience with randomized controlled trials (n = 25).**

| | Overall sample | Ecolectivos team members | External evaluators | p-value[a] |
|---|---|---|---|---|
| | n = 25 | n = 9 | n = 16 | |
| | Median [IQR] | Median [IQR] | Median [IQR] | |
| Knowledge of implementation science[b] | 6 [5, 7] | 5 [4 6] | 7 [6, 8] | 0.23 |
| Experience with RCT design[b] | 8 [5, 8] | 8 [1, 8] | 8 [5, 8] | 0.68 |

[a]Mann Whitney-U Test used to compare medians across the two groups

[b]Knowledge questions were based on a scale of 1–10, with 10 being the highest score

## 2. Recruitment path: *How much extra effort is made to recruit participants over and above what would be used in the usual care setting to engage with patients*?

The median response for recruitment among all respondents was 3 (IQR 2, 4), or equally pragmatic/explanatory. With a relatively balanced perception, evaluators commented on how

**Table 4. PRECIS-2 survey results[a] for overall sample and by Ecolectivos team membership.**

| Domain | Overall sample | | Ecolectivos members | | External evaluators | | p-value[b] |
|---|---|---|---|---|---|---|---|
| | n = 25 | | n = 9 | | n = 16 | | |
| | Median | IQR | Median | IQR | Median | IQR | |
| Eligibility | 4 | 3,4 | 2 | 2, 3 | 4 | 4, 5 | 0.001* |
| *To what extent are the participants in the trial similar to those who would receive this intervention if it was part of usual care?* | | | | | | | |
| Recruitment | 3 | 2, 4 | 2 | 2, 3 | 3 | 3, 4 | 0.08 |
| *How much extra effort is made to recruit participants over and above what would be used in the usual care setting to engage with patients?* | | | | | | | |
| Setting | 4 | 3, 5 | 5 | 3, 5 | 4 | 4, 5 | 0.64 |
| *How different are settings of the trial from the usual care setting?* | | | | | | | |
| Organization | 3 | 2, 4 | 3 | 1,4 | 3 | 2,4 | 0.42 |
| *How different are resources, provider expertise, and organization of care delivery in intervention arm of the trial from those available in usual care?* | | | | | | | |
| Flexibility: *delivery* | 4 | 3, 5 | 3 | 1,5 | 4 | 3,5 | 0.30 |
| *How different is the flexibility in how the intervention is delivered and the flexibility anticipated in usual care?* | | | | | | | |
| Flexibility: *adherence* | 3 | 3, 4 | 3 | 3,5 | 3 | 3,4 | 0.64 |
| *How different is flexibility in how participants are monitored and encouraged to adhere to the intervention from the flexibility anticipated in usual care?* | | | | | | | |
| Follow-up | 2 | 1, 4 | 3 | 2, 3 | 2 | 1,5 | 0.80 |
| *How different is the intensity of measurement and follow-up of participants in the trial from the typical follow-up in usual care?* | | | | | | | |
| Outcome | 3 | 2, 5 | 3 | 1, 4 | 4 | 3,5 | 0.33 |
| *To what extent is the trial's primary outcome directly relevant to participants?* | | | | | | | |
| Analysis | 4 | 2, 5 | 4 | 2, 5 | 4 | 2,5 | 0.85 |
| *To what extent are all data included in the analysis of the primary outcome?* | | | | | | | |

[a] Where 1 = very explanatory to 5 = very pragmatic

[b] Mann Whitney-U Test used to compare medians across the two groups

*Statistically significant at 0.05 level of significance

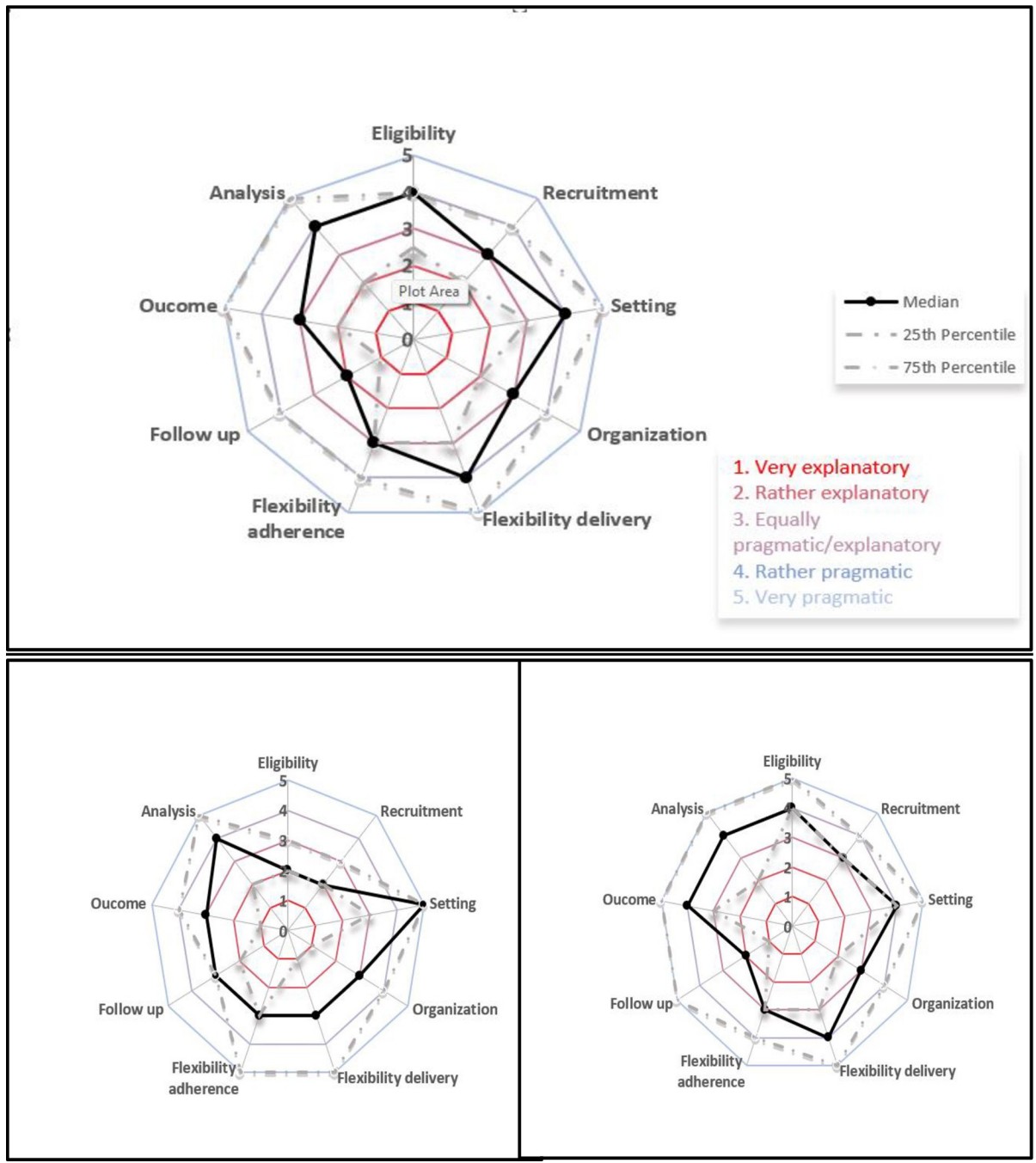

**Fig 1. a.** (top) PRECIS 2 survey response from all participants. **b.** (lower left) PRECIS 2 survey responses from ECOLECTIVOS team members. **c.** (lower right) PRECIS 2 survey responses from external evaluators.

recruitment meets participants "where they are." Leadership from community members through an established Community Advisory Board (CAB) was also referenced as a pragmatic recruitment feature. One evaluator stated, *"the recruitment path, while explanatory in justifying word-of-mouth and informational meetings, is also pragmatic in that it places leadership into community members/stakeholders via CAB input. This represents a pragmatic solution to reaching a dispersed population. I find the approach to be just as pragmatic as it is explanatory for this*

*setting."* The random selection of households for the community-level baseline survey, with village size, distance between villages, and proximity to main roads taken into consideration for stratified sampling, is cited as an explanatory moderating factor. Notably, evaluator commentary spanned several aspects of the recruitment process; for example, recruitment for the recruitment for baseline community survey, which required fieldworkers canvasing neighborhoods door-to-door to administer surveys using google survey maps and census data to randomly select households, was deemed as more explanatory: *"For the formative phase baseline assessment, the door-to-door approach requires extra effort and would not typically be part of usual conditions"*, while recruitment for the biomonitoring randomized controlled trial among women of reproductive age was viewed as more pragmatic, given the methods of recruitment, *"however, for the main trial, recruitment seems to be within usual effort for engagement with this type of project, given the community-based approach, village-level randomization, and widespread existing use of biomass fuel for cooking and burning trash/plastic as a typical means of disposal. Most communities already hold regular meetings or events, so it is expected that approaching people for surveys* [through village meetings] *would not be too extreme of an effort for recruitment."*

### 3. Setting: *How different are the settings of the trial from the usual care setting?*

The median response for setting among all respondents was 4 (IQR 3, 5), or rather pragmatic. Most of the evaluators noted that the intervention will target rural Xinca indigenous communities in a specified region where there is limited waste disposal infrastructure or recycling programs, requiring community-based strategies for plastic risk mitigation. These rural indigenous communities experience similar circumstances as other communities in Guatemala that rely on burning waste for disposal. Evaluators stated that the study setting itself represents the intended audience where outcomes would be utilized, supporting external validity and pragmatic design. One respondent noted that *"[t]he setting in the Jalapa Municipality of Guatemala across eight intervention and eight control villages for the main trial seems to be very pragmatic. This region relies heavily on biomass fuel for cooking and burning trash/plastic as a usual means of disposal. The villages will capture a range of settings and natural conditions, rather than just a single, specialized site. The in-home assessments of exposures and biomarkers of exposure are also all occurring in very real-world conditions."* Of note, one evaluator pointed out that the longstanding relationship between researchers and the community may create results that deviate from "usual" community intervention settings, *"as with any RCT, it is rather site-specific, and it's a single site study in a sense. It's also a study with a long relationship with the investigator team, so there may be more compliance to the intervention, higher acceptability than in other real-world settings."*

### 4. Organization: *How different are the resources, provider expertise, and the organization of care delivery in the intervention arm of the trial from those available in usual care?*

The median response for organization among all respondents was 3 (IQR 2, 4) or equally pragmatic/explanatory. Evaluator feedback highlighted several organizational aspects of the intervention, including its feasibility, resources required, and involvement of community members. The intervention's combination of scientific and community expertise generated a near split in opinion between pragmatic vs explanatory features. Although the study involves external expertise and resources to deliver the intervention, it involves community members in developing interventions, which adds pragmatism. Overall, this was assessed as a balanced

approach. One evaluator scored this domain as "rather explanatory" and commented on the complexity of the organizational challenges leading to a more explanatory assessment, given that external resources are required: *"[T]he trial requires the efforts of many fieldworkers and external investigators to pull off the intervention, none of which are part of what would normally happen."* Discussion around more explanatory features of the trial also included that training and resources for the 12-week educational sessions are intensive, requiring specific expertise and external resources, and *"the intervention is not "standard" care or part of existing community-based services."* The involvement of fieldworkers to improve adherence is also noted by some as shifting their trial rating towards explanatory. On the other hand, since the intervention takes place within the community setting, other evaluators found it practical for the targeted population, as evidenced by features such as use of community centers for workshops, and the role of key informants and village champions to support community conditions for behavior change. Several evaluators pointed out the likely scalability of the intervention, which may be delivered similarly under real-world conditions, increasing its pragmatic usefulness.

### 5. Flexibility of experimental intervention–delivery: *How different is the flexibility in how the intervention is delivered and the flexibility anticipated in usual care?*

The median response for flexibility of the experimental intervention in delivery among all respondents was 4 (IQR 3, 5), or rather pragmatic. Several respondents noted the formative phase as a strength, with one stating: *"[T]he formative [phase] followed by the RCT will allow for a balance between the pragmatism of the intervention (and allowing the community to drive its development) while providing scientific evidence for the benefit of the selected intervention in the later stages."* In provided rationale, evaluators noted that the protocol is adaptable and responsive to community input, allowing participants to choose intervention approaches that appeal to them and address their specific barriers: *"It is very clearly mentioned as to what modules are fixed and which are flexible. It serves the purpose of the project as well as needs of the community for the skill to be learned. It's a win-win situation."* Since participants play an active role in generating intervention ideas, this was seen to favor pragmatic design and support external validity, or *"the details of implementation and intervention are flexible to community members' needs and decision-making. This creates an intervention that should proxy what happens under usual circumstances."* The multi-prong approach with evidence-based strategies, the formation and structure of community working groups, and community education sessions are noted to be externally driven and less flexible. Evaluators identified features that seemingly strike a balance between a pragmatic and a structured, more explanatory delivery plan.

### 6. Flexibility of experimental intervention–adherence: *How different is flexibility in how participants are monitored and encouraged to adhere to the intervention from the flexibility anticipated in usual care?*

The median response for flexibility of the experimental intervention in adherence among all respondents was 3 (IQR 3, 4), or equally pragmatic/explanatory. Evaluators noted that there are established groups in place, such as village champions and trained fieldworkers "available to resolve bottlenecks", which are intended to increase adherence. However, several respondents felt that details about specific measures of adherence, such as "*how many times study staff will reach out to participants to get them to come to the sessions*" or *"field workers will check on participants, but there's no way to know what the participants are doing when not being watched"* would have further clarified the pragmatism of protocol design. With the last four weeks of the educational intervention being participant-directed, evaluators noted this factor

as likely to increase flexibility and community buy-in, with one stating *"the fact that the participants get a chance to provide their input already increases their buy-in which will promote adherence to the study. While it is good to have strict protocols, it is important that you are incorporating room for flexibility with the modules."*

## 7. Follow-up: *How different is the intensity of measurement and follow-up of participants in the trial from the typical follow-up in usual care?*

The median response for follow-up among all respondents was 2 (IQR 1, 4), or rather explanatory. Respondents focused attention on the follow-up of the biomonitoring group of 400 women who would be visited three times in their homes to assess exposure to air pollution. Evaluators noted that the regular follow-up surveys and biomarker assessments require trained data collectors, described by most (52%) to be very or rather explanatory. One respondent stated, *"this trial requires regular assessments of 200 intervention and 200 control participants across 16 villages for monitoring exposure and urinary biomarkers of exposure at baseline, 4 months, and 12 months. There is also scheduled visits for plastic waste collections at baseline and 4 months. This is fairly rigorous for measurement of trial outcomes and requires extensive data collection."* However, two respondents thought that the trial was pragmatic. One stated that the trial was "very pragmatic" since a "*12-month post-intervention measure could be insufficient*" if adherence to the intervention geared at "not burning plastic" is not followed. Another evaluator stated that the trial was "rather pragmatic" because follow-up after the educational intervention was infrequent, and exhibited *"minimal follow-up visits, data collection, and dissemination performed within the community group setting. This represents close to usual follow-up without any demanding expectations for participants."*

## 8. Primary outcome: *To what extent is the trial's primary outcome directly relevant to participants?*

The median response for the primary outcome among all respondents was 3 (IQR 2, 5), or equally pragmatic/explanatory. The biomarker sampling procedures, which measure exposure to air pollutants, including pollutants from plastic waste burning, were by some, deemed as tangible features of trial outcomes, and "*directly relate to participants' exposure burden.*" However, it was pointed out that participants may not find these data immediately relevant *a priori* if they are disconnected from the health effects associated with household air pollution. Others felt that the importance of intervention strategies and their direct relevance to participants were pragmatic in that the "*educational intervention would likely hold significance to participants*" and "*villagers expressed their desire to manage plastic in ways other than burning*". Feedback indicated that certain outcomes may be less relevant to participants, but they are important for evaluating the effectiveness of the intervention and research. One respondent summarized, *"while the primary outcomes are important to participants' health, I don't think these outcomes will be what would entice participants to make changes or continue those changes outside of the study. No one is going to think "I am going to recycle this plastic instead of burn it because it will keep my bisphenol levels stable." People make changes because of the importance for their everyday life circumstances (finances, ease, and so on). I think the secondary outcomes [health-related quality of life and general self-efficacy] are more explanatory than the primary ones [air pollution exposure and urinary biomarkers of exposure]."*

### 9. Primary analysis: *To what extent are all data included in the analysis of the primary outcome?*

The median response for the primary analysis among all respondents was 4 (IQR 2, 5), or rather pragmatic. The intention-to-treat analysis of quantitative biomarkers, which *"does not have many exclusions of data for primary analysis,"* led some respondents to comment that the study was more explanatory than pragmatic. A strength of the study was in the *"mixed methods approach that will combine data from multiple sources to evaluate the intervention."* Overall, the feedback suggests that the data analysis plan, including the handling of missing data, appears to be pragmatic and well-considered. Many stated that the intention to include all available data, even for participants who may drop out or become pregnant during the study, reinforces the perception of a more pragmatic approach. This was well summarized in one statement, *"there is no systematic exclusion of data detailed in the protocol; the intervention protocol details how any missing data will be addressed during statistical evaluation of all data collected. Data analysis is well defined and pragmatic."*

## Discussion

Using the PRECIS-2 tool to evaluate the Ecolectivos study protocol, evaluators scored the trial design as more pragmatic than explanatory, with eight of the nine evaluated domains ranked >3 (equally pragmatic/explanatory) on a Likert scale of 1–5. Only the *follow-up* domain was assessed to be more explanatory in nature (median, 2 or rather explanatory). The broad consensus in scoring across these eight domains demonstrate consistency across all independent evaluators, aligning Ecolectivos investigators with external evaluators in their assessment of trial design. *Eligibility* was the only domain that showed a statistically significant difference in rankings between Ecolectivos investigators (median, 2 or rather explanatory) and external evaluators (median, 4 or rather pragmatic) (*p*-value <0.001). The differing perceptions of pragmatism as related to *eligibility* may be explained by conflicting ideas of the perceived intended audience. As blinding of individuals is not possible in a behavior change intervention, village-level cluster randomization was considered to be a suitably pragmatic design. The selection of the sample is noted as representative of the Jalapa, Guatemala population, which some considered pragmatic as it increases the likelihood of successful implementation of the intervention in that specific context. However, others questioned whether the formative phase and the related contextual intervention adaptation would impede the trial's ability to provide insights that could be generalized to different populations or settings, in Guatemala or in other low-resource settings, where plastic waste is burned as a primary means of waste disposal. We counter this argument with evidence from our earlier work, which used the COM-B/TDF model to develop a gas stove intervention in rural Guatemala [33] that was later adapted for use in a multi-country RCT among 3,2000 pregnant women and their families [34]. This demonstrates that ubiquitous problems, like access to clean energy and proper waste disposal, have generalizable solutions if behavioral change is carefully assessed, with intervention strategies that map onto these behaviors. Using an established framework, like COM-B/TDF, to systematically inform intervention design that addresses barriers to plastic waste management in one context, may have applicability in other low-resource countries facing this same problem.

Understanding how specific intervention strategies are further adapted to the Jalapa setting, following our initial theory-informed approach to developing the specific community intervention components included in Ecolectivos, is the goal of this study and builds on other environmental intervention work in rural Guatemala [15, 17, 26]. In the Ecolectivos study, we will evaluate to what extent this approach can lead to reduced plastic waste burning in the participating communities. Each of the eight intervention communities will receive eight weeks of a

standard educational workshop, after which participants will spend four weeks proposing vil-lage-level strategies that they want to adopt for the ensuing nine months. While there will be a set number of activities that can be chosen (e.g., recycling, composting, community clean-up), each village will adapt the activities to meet their needs. We anticipate from our pilot work that some community members will be highly engaged, while others will be less interested in pur-suing activities. Thus, adaptation of the selected activities may vary by community, dependent upon contextual factors that may limit the generalizability of the strategies to other settings beyond the region.

Implementation science seeks to promote the uptake of evidence-based practices with the goal of improving health outcomes [35]. While much of the science has focused on implement-ing change in clinical settings, complex environmental health problems, like plastic waste burn-ing in low-resource community settings, are incredibly challenging to address at many levels [36]. Ecolectivos aims to enable community-driven alternatives to plastic waste, through strate-gies like recycling, community clean-ups, and refusing to use plastic, which may lead to visibly tangible environmental improvements at the household and village level. However, as pointed out by survey respondents, study participants may not perceive health benefits from reduced exposures to air pollution from waste burning, and proposed study measurements (personal exposures to air pollution and urinary biomarkers of exposure) may not be sufficient to moti-vate participants to make behavioral changes even if there were reductions in exposure. This has been found in household air pollution research, where the irritation from smoke exposure (e.g., red irritated eyes, cough), or observable soot on kitchen walls, are cited as motivators to reduce exposure, compared to health outcomes (e.g., pneumonia, high blood pressure) [37].

Globally, air pollution (both indoor and outdoor) represents the greatest environmental risk factor for poor health outcomes, contributing to an estimated seven million premature deaths in 2019 alone [38]. An estimated two billion people burn waste as a principal means of disposal globally [39]. The extent to which the burning of household waste contributes to ambient and household air pollution is a considerable, but poorly quantified problem, that needs to be addressed. But *how*? Although extensive system and policy changes are necessary for reducing the burden of plastic waste accumulation in the environment, the move towards a circular plastics economy will need to be mediated through behaviors of the global populace [18]. In other words, a closed-loop system where the flow of plastics is repurposed, reused, recycled, and recovered may have the potential to address the mounting burden of single-use plastic, a linear economy from production to waste, and the resulting cumulative plastic waste in the environment [40]. While technological improvements and changes to infrastructure are necessary components of the bigger picture, these alone will not be enough to remedy the global plastic waste burden. In concert with larger systemic efforts to mitigate the harms of plastic technologies, global populations still must interact with these technologies and larger systems to facilitate any environmental health advantage [18].

## Limitations and strengths

Although we had a lower participation rate, our response rate was higher than many online surveys. The consequent small sample size (n = 25) can affect overall generalizability. Further, we had greater participation from Ecolectivos investigators (64%) than from external reviewers (35%), which is not surprising; thus, response bias may be a concern for interpretation of find-ings. Because we found a strong consensus on the pragmatic and explanatory features of the study across these two groups, this enables a broadened perspective on whether the implemen-tation protocol will serve its intended purpose with consideration toward external validity. Of note is that although the median self-rated knowledge and experience with implementation

science was higher among external evaluators (7/10) than among Ecolectivos investigators (5/10), results were similar. Other limitations were the requirement to be an English speaker, which meant that some members of our Guatemalan team could not evaluate the study, and that the online survey only included short-answer responses, which limited the richness of the data for qualitative analysis. Study strengths include the incorporation of assessments by implementation science experts external to the Ecolectivos research collaborative, which may serve to offset biases had the protocol been evaluated only by Ecolectivos investigators, and the uniqueness of the topic in the implementation science intervention space. Our findings contribute to the growing body of literature tying community environmental health interventions to the rigor of implementation science.

## Conclusions

While much of the focus of the use of PRECIS-2 has been aimed at the evaluation of clinical trials [41], our study calls attention to the need to evaluate research study protocols before study implementation to characterize complex, community-based environmental health intervention studies along the pragmatic/explanatory continuum using the PRECIS-2 tool. Environmental health research generates valuable insights and evidence-based practices to improve public health; however, the translation of research findings into effective, real-world interventions is often challenging due to the myriad and complex drivers affecting intervention research in community settings. Our protocol was viewed as more pragmatic than explanatory, providing evidence to support the goal of the Ecolectivos study, which is to address greater research utility through a more pragmatic trial design for the reduction of plastic waste burning and plastics use in communities without waste removal resources. Through enhanced investigator assessment and communication regarding intervention protocols, resource efficiency and study design may be fine-tuned through implementation science tools to ensure effective program implementation that meets the stated goals of a proposed intervention.

## Supporting information

**S1 Checklist. STROBE statement—Checklist of items that should be included in reports of *cross-sectional studies*.**
(DOCX)

## Author Contributions

**Conceptualization:** Lisa M. Thompson.

**Data curation:** Amy E. Lovvorn, Mayari Hengstermann-Artiga.

**Formal analysis:** Hina Raheel, Melinda Higgins.

**Methodology:** Lisa M. Thompson, Annalyse Ferguson.

**Supervision:** Lisa M. Thompson.

**Visualization:** Hina Raheel, Melinda Higgins.

**Writing – original draft:** Lisa M. Thompson, Annalyse Ferguson.

**Writing – review & editing:** Lisa M. Thompson, Annalyse Ferguson, Hina Raheel, Amy E. Lovvorn, Mayari Hengstermann-Artiga, Maria Renee Lopez, Melinda Higgins, Eri Saikawa, Margaret A. Handley.

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
