## [Decision Letter · Decision Letter 0]

30 Sep 2024

PONE-D-24-25892PRECIS-2 used as an implementation science tool for global environmental health:

A cross-sectional evaluation of the Ecolectivos study protocol to reduce burning of household plastic waste in rural GuatemalaPLOS ONE

Dear Dr. Thompson,

Thank you for submitting your manuscript to PLOS ONE. After careful consideration, we feel that it has merit but does not fully meet PLOS ONE’s publication criteria as it currently stands. Therefore, we invite you to submit a revised version of the manuscript that addresses the points raised during the review process.

Thank you for your valuable study.

I genuinely enjoyed reading your work, and I can see the thought and endeavour behind it. However, I believe that some of the comments will need more time to address. The idea is very good, and I appreciate your intention, but a bit more work will be needed.

I will list all comments along with my interpretation and suggestions for improvement:

1) Please double-check grammar (e.g. repetition and verb tense);

2) Please double-check refs (e.g. punctuation and formatting);

3) The Abstract is clear and well-structured, but certain refinements are needed:

- First, the transition between terms, specifically the connection between RCTs and the PRECIS-2 tool, is abrupt. The link between the efficacy of the tool and its application could be more clearly outlined. A simple addition explaining how the tool applies to RCTs and enhances their design in real-world contexts can resolve this. For example, ;

- The debate on "RCTs being underutilised" would be more suitable as an in-depth part of the Introduction rather than the Abstract;

- If the authors prefer to elaborate on RCTs to provide a stronger rationale, consider making a clear link between RCTs and the intervention. If the intention is to argue against other, this argument could be reframed within the Introduction;

- Please clarify the real-world application of the tool, including its background and the framework supporting its use;

- The Methods section could begin with an overview of the overall characteristics, followed by an introduction to the PRECIS-2 framework. This will refine readability while also adhering to the guidelines;

- Some parts of the Results section need explanation re. their importance to the findings;

- Please Work on a stronger conclusion debating the applications and implications for environmental health, for example;

- I understand that the debate on RCTs at the beggining of Introduction was provided in similar studies (e.g. 10.1136/bmj.h2147), but it may no be appropriate for a broader audience in this current form;

4) The authors report the ID (i.e. NCT05130632) but provided the STROBE checklist, also mentioned as observational. Although this isn't wrong, some concerns were noted:

- Neither the STROBE nor CONSORT guidelines appear to be adhered to in the ms. The - important - items based on the study design could be addressed to ensure compliance. I suggest a thorough check of the applicable guidelines (i.e. STROBE or CONSORT) and a careful revision of both the Abstract and the ms;

- Insufficient detail regarding the sample, design, and background, in both Abstract and text, making it unclear what the variables, outcomes, and implications are;

- In its conciseness, the Background section describes the intentions. There are interesting explanations re. the importance of RCTs over other designs (e.g. "This is especially relevant in community-based environmental studies"). This construction of argument is important for setting the tone for both the readers and a broader audience. However, despite the commendable attention to detail, the authors appear to be aware of the limitations in clearly defining their study.   In the Study Details section, the context, rationale, and gaps are somewhat overextended (e.g. "Household air pollution from solid fuel combustion is a major environmental risk factor in low- and middle-income countries, accounting for an estimated 2.6 million deaths annually (World Health Organization, 2018). The contribution of plastic waste incineration in household fires has not been quantified. This is problematic for countries like Guatemala, where 71% of households burn waste as a primary means of disposal (Government of the Republic of Guatemala, 2019)."). While informative, this introduction of the problem is not as clear as it could be. I highly suggest reshaping the Introduction to improve flow and readability, focusing on: a) background, b) rationale, c) main gaps in previous studies, d) the motivation behind this study, e) its significance, and f) the hypotheses and objectives;

- The refs. need to be refined and integrated more thoroughly throughout the text, as there are several statements without citations. For example, more information on the validity, reliability, subjectivity, or generalisation, especially re, the proposed RCT—could be strengthened with additional references (e.g. 10.1186/s13063-023-07313-0; 10.1016/j.jclinepi.2020.03.<wbr style="color: rgb(34, 34, 34); font-family: Arial, Helvetica, sans-serif; font-size: small;" />027);

- Please streamline the Method section, emphasising why specific aspects or markers are important. Currently, some details are repeated from other studies rather than focusing on improving the clarity of the hypotheses and objectives;

5) Both the Methods and Results sections lack the necessary detail, making it difficult to evaluate the manuscript in its current form. I'll list a few questions for the authors - especially based on the guidelines and necessity for refine scientific communication;

- Why not begin by describing a key aspect of your study—the sample?;

- Where are the power calculations or sample size determinations? This is *essential*. Additionally, consider addressing the differences in the invitee ratios;

- Please ensure adhering to the guidelines when reporting the data storage and handling;

- What are the confounding factors and possible intervening variables, and how were they managed? Consider response bias as a primary issue;

- Please include details on the psychometric properties of the tool, including the scoring and rating process;

- Was the tool validated?;

- Why is no mention of the specific tests, their assumptions, parameters and explanations? Although Mann-Whitney is mentioned, no explanation is provided. Please consider detailing all parameters, including effect sizes and CIs;

- A detailed section for the qualitative part it essential;

- Explain the modifications to the study?;

Overall, the Methods and Results sections need to be refined, particularly as they do not currently adhere to the relevant guidelines;

6) There seems to be a misunderstanding, as the Results section includes aspects that should belong in the Methods section;

- More detail on eligibility, analyses, and scoring methods are relevant;

- How the data were collected or measured, details of survey questions, and the scales used for evaluation could be provided in Methods;

- Correlations are also advisable - as well as potential regression analyses;

7) Based on this, it is difficult to fully assess the data without proper alignment with the guidelines;

8) The formative phase baseline recruitment, involving door-to-door approaches, was considered more explanatory due to the additional effort required, while recruitment for the main trial was viewed as pragmatic, with village-level randomisation and community-based approaches that aligned with usual community engagement. Please elaborate;

9) The authors should consider checking their data again. For example, in Tables:

10) There is the need to expand a few more demographics, specifically the ones that could serve as confounding factors as IQ, education or previous experiences. Also in Table 2, there are no descriptions of additional tests (e.g. chi-squared). Also check of how these influence the other variables;

- In Table 3, please expand including effects, CIs and check outliers. Considering the small sample, the IQRs can be double-checked;

- In Table 4, the first two domains exhibited significant results after correcting the data. Even without adjusting, the results for recruitment were also significant. I'd highly suggest the authors to double-check;

11) The dispersion of data (i.e. IQR) indicates both the possibility of influence of pre-existing knowledge, as that sample could be biased, i.e. if sample were larger, then probably outcomes would be different. This can be better observed;

We look forward to receiving your revised manuscript.

Kind regards,

Thiago P. Fernandes, PhD

Academic Editor

PLOS ONE

Journal requirements: 1. When submitting your revision, we need you to address these additional requirements.Please ensure that your manuscript meets PLOS ONE's style requirements, including those for file naming. The PLOS ONE style templates can be found at https://journals.plos.org/plosone/s/file?id=wjVg/PLOSOne_formatting_sample_main_body.pdf and https://journals.plos.org/plosone/s/file?id=ba62/PLOSOne_formatting_sample_title_authors_affiliations.pdf 2. Please amend either the title on the online submission form (via Edit Submission) or the title in the manuscript so that they are identical. 3. Your abstract cannot contain citations. Please only include citations in the body text of the manuscript, and ensure that they remain in ascending numerical order on first mention. 4. Please include a caption for figure 1. 5. Thank you for stating the following financial disclosure:  [The Ecolectivos Trial is supported by the National Institute of Environmental Health Sciences (NIEHS) of the National Institutes of Health under Award Number R01ES032009. The content is solely the responsibility of the authors and does not necessarily represent the official views of the National Institutes of Health].  Please state what role the funders took in the study.  If the funders had no role, please state: ""The funders had no role in study design, data collection and analysis, decision to publish, or preparation of the manuscript."" If this statement is not correct you must amend it as needed. Please include this amended Role of Funder statement in your cover letter; we will change the online submission form on your behalf. 6. In the online submission form, you indicated that [The datasets used and/or analysed during the current study are available from the corresponding author on reasonable request.]. All PLOS journals now require all data underlying the findings described in their manuscript to be freely available to other researchers, either 1. In a public repository, 2. Within the manuscript itself, or 3. Uploaded as supplementary information.This policy applies to all data except where public deposition would breach compliance with the protocol approved by your research ethics board. If your data cannot be made publicly available for ethical or legal reasons (e.g., public availability would compromise patient privacy), please explain your reasons on resubmission and your exemption request will be escalated for approval.  7. Your ethics statement should only appear in the Methods section of your manuscript. If your ethics statement is written in any section besides the Methods, please move it to the Methods section and delete it from any other section. Please ensure that your ethics statement is included in your manuscript, as the ethics statement entered into the online submission form will not be published alongside your manuscript.  8. Please include captions for your Supporting Information files at the end of your manuscript, and update any in-text citations to match accordingly. Please see our Supporting Information guidelines for more information: http://journals.plos.org/plosone/s/supporting-information. 

Reviewers' comments:

Reviewer's Responses to Questions

**Comments to the Author**

1. Is the manuscript technically sound, and do the data support the conclusions?

Reviewer #1: Yes

Reviewer #2: Yes

Reviewer #3: Yes

2. Has the statistical analysis been performed appropriately and rigorously? 

Reviewer #1: Yes

Reviewer #2: Yes

Reviewer #3: Yes

3. Have the authors made all data underlying the findings in their manuscript fully available?

Reviewer #1: Yes

Reviewer #2: Yes

Reviewer #3: Yes

4. Is the manuscript presented in an intelligible fashion and written in standard English?

Reviewer #1: Yes

Reviewer #2: Yes

Reviewer #3: Yes

5. Review Comments to the Author

Reviewer #1: PONE-D-24-25892

"PRECIS-2 used as an implementation science tool for global environmental health: A cross-sectional evaluation of the Ecolectivos study protocol to reduce burning of household plastic waste in rural Guatemala"

This study evaluates the Ecolectivos research protocol, which aims to reduce plastic waste burning in rural Guatemalan communities, using the Pragmatic Explanatory Continuum Indicator Summary (PRECIS-2) framework. Researchers and study partners assessed the protocol across nine domains, finding that four domains were clearly pragmatic, four were mixed, and one was mainly explanatory. Results highlighting the study's practical aspects and contribute to the integration of implementation science in environmental health research. The manuscript is well-written and methods are very clear. Comments are below.

• I was excited to see a manuscript so clearly describing implementation science methodology in an environmental health context - I look forward to seeing future results from this project as it progresses. For readers (especially if only vaguely familiar), it could be helpful to define implementation science as a basic concept and further emphasize its importance in environmental health intervention work (in the introduction). Could just be a sentence or two.

[page 7, line ~107] “Theory and evidence-based behavior change research are necessary to guide these interventions.”

• Add citation? I see you have relevant references. Or would like to see citations here regarding relevant behavior change research concepts for readers looking for a basic overview if unfamiliar.

• Beyond this, a few citations supporting the importance and understanding of the community-engaged research umbrella (which includes participatory action research, CBPR) would strengthen the introduction. The work you are doing is very important, especially when considering it in an EJ context – you mention EJ once in the background. Unless you define EJ further and/or include explicit contextual EJ examples in the study community, I might consider removing the term EJ. However, there is an opportunity to further emphasize community-engagement (Minkler & Wallerstein, 2008) as this manuscript describes an important first step of a larger community-based implementation research plan.

[page 23 line ~473] “Other limitations were the requirement to be an English speaker, which means that some members of our Guatemalan team could not evaluate the study, and that the on-line survey only included short answer responses, which limited the richness of the data for qualitative analysis.”

• Could you add details about any future study steps that will address bilingual engagement with Spanish-speaking study team members, if at all? Will you access a translation service? Translator? Is there an opportunity to report back the results from this manuscript in Spanish to team members after the fact? If you are at all framing the overall study in an EJ context and/or using community-engaged approaches, this must be addressed.

• To consider: it would be interesting to see a brief section/paragraph reflecting on your partnership within Ecolectivos (Guatemalan team partners) thus far. Could be an opportunity to discuss the bullet point above.

Reviewer #2: Dear Author,

The paper is written in a very good quality of structure and clarity of the purpose of the study and interpretation of the findings. It is very important because it breakdown different domain of quasi experiments to measure its pragmatism or explanatory level. This could be a good evaluation step in developing action research protocols. I only have a few suggestions that might be considered to improve the paper.

1. the author could state her/his position as developer of the Ecolectivos protocols or outside parties of the Ecolectivos study? This could affect the interpretation of the findings by the author. and how the author address potential bias because of the role in Ecolectivos

2. It would be valuable for readers if the paper also present example of other intervention (as explained in the discussion section) and compare it similarity of different approach with Ecolectivos for each domain which might increase or reduce explanatory/pragmatic level.

3. I would like to see recommendations on what should be improved/important to be redesigned, or what domain actually the correct (expected) approach for being pragmatic or explanatory in this study or other environmental health intervention study?

Overall, the paper is very insightful in the field of intervention studies

Thank you

Reviewer #3: This manuscript outlines the use of the implementation science tool, PRECIS-2 to evaluate the RCT protocol of the Ecolectivos Study. This study aims to implement interventions to reduce plastic waste burning in rural Guatemala. This paper is a great example of using implementation science in environmental health settings and will hopefully move the field to be more rigorous in their implementation protocols!

The methods and results sections are straightforward and well laid out. I felt that there was some slight confusion in the introduction and conclusion around whether it was the Ecolectivos protocol or the PRECIS-2 tool itself that was being evaluated. In particular the last section of the abstract "Conclusion" makes it sound like the tool itself was being evaluated rather than used to evaluate the Ecolectivos protocol. I also recommend switching the order of the introduction to first introduce the Ecolectivos study and its goals and then move into the use of PRECIS-2 and implementation science tools to evaluate RCTs.

Minor comments:

Line 161 – how did you identify external participants as experts in their field?

Line 418-421 - had to read this sentence multiple times to understand the meaning - consider rewording.

Lines 421-430 feel out of place in the discussion and should be moved to the introduction as part of the description of the Ecolectivos study

6. PLOS authors have the option to publish the peer review history of their article (what does this mean?). If published, this will include your full peer review and any attached files.

Reviewer #1: No

Reviewer #2: **Yes: **Ni Made Utami Dwipayanti

Reviewer #3: No

---

## [Author Response · Author response to Decision Letter 0]

21 Nov 2024

RESPONSE TO REVIEWERS: 

1) Please double-check grammar (e.g. repetition and verb tense)

Reviewer Response: Thank you we have we have reviewed and modified the grammar throughout the text. 

2) Please double-check refs (e.g. punctuation and formatting)

a. The refs. need to be refined and integrated more thoroughly throughout the text, as there are several statements without citations. For example, more information on the validity, reliability, subjectivity, or generalisation, especially re, the proposed RCT—could be strengthened with additional references (e.g. 10.1186/s13063-023-07313-0; 10.1016/j.jclinepi.2020.03.027).

Reviewer Response: We have fixed the references and added references where they are needed throughout the text. 

ABSTRACT REVIEWER COMMENTS

3) The Abstract is clear and well-structured, but certain refinements are needed:

a. First, the transition between terms, specifically the connection between RCTs and the PRECIS-2 tool, is abrupt. The link between the efficacy of the tool and its application could be more clearly outlined. A simple addition explaining how the tool applies to RCTs and enhances their design in real-world contexts can resolve this. 

Reviewer Response: We were constrained by the guidelines not to exceed 300 words but have tried to address (3 a-g) all of which pertain to the abstract, to the best of our ability. The abstract is now 450 words.

We have re-written the first few sentences to clarify the relationship between efficacy and effectiveness and the function PRECIS plays in the evaluation of pragmatism in trial designs. We hope this resolves the reviewer’s concerns. 

b. For example, the debate on "RCTs being underutilised" would be more suitable as an in-depth part of the Introduction rather than the Abstract.

Reviewer Response: We feel that is an important part of the argument, that RCTS are underutilized in environmental health studies, and thus should be included that in the abstract, but we elaborated more on this in the introduction as the reviewer suggested. 

c. If the authors prefer to elaborate on RCTs to provide a stronger rationale, consider making a clear link between RCTs and the intervention. If the intention is to argue against other, this argument could be reframed within the Introduction.

Reviewer Response: We are not sure if we interpreted the reviewer’s comment correctly, but the purpose of the PRECIS-2 is to evaluate the degree of pragmatism of an RCT. We used this tool to evaluate our protocol that focuses on a randomized intervention delivered in a community setting to address an environmental health concern. We tried to clarify that more in the abstract. We are not clear about the second sentence and what “other” refers to. 

d. Please clarify the real-world application of the tool, including its background and the framework supporting its use.

Reviewer Response: We have clarified this in lines 3-6 of the abstract. 

e. The Methods section could begin with an overview of the overall characteristics, followed by an introduction to the PRECIS-2 framework. This will refine readability while also adhering to the guidelines.

Reviewer Response: We moved the sample and study characteristics to the beginning of the methods section and then described the PRECIS-2 tool. 

f. Some parts of the Results section need explanation re. their importance to the findings.

Reviewer Response: We have strengthened the importance of the findings in lines 30-33. 

g. Please work on a stronger conclusion debating the applications and implications for environmental health, for example-- I understand that the debate on RCTs at the beginning of Introduction was provided in similar studies (e.g. 10.1136/bmj.h2147), but it may not be appropriate for a broader audience in this current form.

Reviewer Response: We have rewritten the conclusion. 

4) The authors report the ID (i.e. NCT05130632) but provided the STROBE checklist, also mentioned as observational. Although this isn't wrong, some concerns were noted:

a. Neither the STROBE nor CONSORT guidelines appear to be adhered to in the ms. The - important - items based on the study design could be addressed to ensure compliance. I suggest a thorough check of the applicable guidelines (i.e. STROBE or CONSORT) and a careful revision of both the Abstract and the ms.

Reviewer Response: We included the STROBE guidelines to describe this cross-sectional (observational) study. We have reviewed the criteria and clarified in the manuscript. Some are N/A (e.g. flow diagram). We provided the NCT trial number because this cross-sectional study is evaluating the protocol for a randomized controlled trial. The NCT provides more detail about the study that is being evaluated. We also include a brief overview of the trial in the introduction to introduce the reader to the study. For that reason, we only include the STROBE guidelines only for the present manuscript. 

b. Insufficient detail regarding the sample, design, and background, in both Abstract and text, making it unclear what the variables, outcomes, and implications are.

Reviewer response: We have added details about the sample, design and background to the abstract as described above in 1). We have also added details about sample and design in the methods as described in 5). We have elaborated on the Background as described in 4c). We hope that this is sufficient to address these concerns.

c. In its conciseness, the Background section describes the intentions. There are interesting explanations re. the importance of RCTs over other designs (e.g. "This is especially relevant in community-based environmental studies"). This construction of argument is important for setting the tone for both the readers and a broader audience. However, despite the commendable attention to detail, the authors appear to be aware of the limitations in clearly defining their study. In the Study Details section, the context, rationale, and gaps are somewhat overextended (e.g. "Household air pollution from solid fuel combustion is a major environmental risk factor in low- and middle-income countries, accounting for an estimated 2.6 million deaths annually (World Health Organization, 2018). The contribution of plastic waste incineration in household fires has not been quantified. This is problematic for countries like Guatemala, where 71% of households burn waste as a primary means of disposal (Government of the Republic of Guatemala, 2019)."). While informative, this introduction of the problem is not as clear as it could be. I highly suggest reshaping the Introduction to improve flow and readability, focusing on: a) background, b) rationale, c) main gaps in previous studies, d) the motivation behind this study, e) its significance, and f) the hypotheses and objectives.

Reviewer response: Thank you for highlighting our intended use of the PRECIS-2 to evaluate a protocol for a community-based intervention study to address an environmental problem. We reorganized the introduction as suggested, explicitly stating “rationale” “gaps” “motivation” significance” and “objectives”. We are not testing hypotheses in this exploratory study. We removed the statement about the lack of global waste management in the introduction but left it in the discussion, to contextualize the comments made by the evaluators who reviewed the protocol using the PRECIS-2 tool. We have looked throughout the paper for the statements (e.g. "Household air pollution from solid fuel combustion is a major environmental risk factor in low- and middle-income countries, accounting for an estimated 2.6 million deaths annually (World Health Organization, 2018). The contribution of plastic waste incineration in household fires has not been quantified. This is problematic for countries like Guatemala, where 71% of households burn waste as a primary means of disposal (Government of the Republic of Guatemala, 2019).") but realize that these statements refer to the actual RCT that we are conducting which is not part of the manuscript under review. Those statement are background for the larger study described in the protocol, therefore we did not address this reviewer comment.

d. Please streamline the Method section, emphasising why specific aspects or markers are important. Currently, some details are repeated from other studies rather than focusing on improving the clarity of the hypotheses and objectives.

Reviewer response: Methods have been streamlined and describe the present study. We described the sample, the sampling method, and the survey tool that was used to evaluate the protocol. We also describe the demographic variables that were collected. We hope that this improves the clarity of the objectives of this manuscript. 

5) Both the Methods and Results sections lack the necessary detail, making it difficult to evaluate the manuscript in its current form. I'll list a few questions for the authors - especially based on the guidelines and necessity for refine scientific communication.

a. Why not begin by describing a key aspect of your study—the sample?

Reviewer response: We have added a description of our intended sample at the beginning of the methods section and the actual sample at the beginning of the results section. 

b. Where are the power calculations or sample size determinations? This is *essential*. Additionally, consider addressing the differences in the invitee ratios

Reviewer response: While we understand the importance of power analyses and sample size calculations in studies that are testing hypotheses, this was an exploratory study with 25 participants who reviewed our protocol. We have clarified that we did not conduct a power analysis in the data analysis section, lines 234-235.

We had already addressed the differences in the invitee ratios in the discussion but added a clarifying phrase in lines 211-215.

c. Please ensure adhering to the guidelines when reporting the data storage and handling

Reviewer response: We have added this verbiage to lines 182-184.

d. What are the confounding factors and possible intervening variables, and how were they managed? Consider response bias as a primary issue

Reviewer response: We wish that we had a larger sample size to examine potential demographic covariates, but with 25 participants we could not. We did instead attempt to examine differences based on external evaluators versus Ecolectivos investigators, with the caveat that there would be potential bias as elaborated upon in the results (lines 486-488). 

e. Please include details on the psychometric properties of the tool, including the scoring and rating process

Reviewer response: The PRECIS-2 tool has not been evaluated as rigorously as one might when developing scales to measure latent variables for psychological instruments (e.g., stress, anxiety) that use underlying measurement theory. In (f) we describe the validity and reliability testing that has been undertaken by the developers of the instrument. 

f. Was the tool validated?

Reviewer response: Yes, the PRECIS-2 tool has been validated, and we have emphasized that in several areas and explained the validation in lines 178-180.

g. Why is no mention of the specific tests, their assumptions, parameters and explanations? Although Mann-Whitney is mentioned, no explanation is provided. Please consider detailing all parameters, including effect sizes and Cis

Reviewer response: In lines 176-179 of our original draft we stated “Variable distributions were evaluated to assess whether assumptions of normality were met; nonparametric tests were used based on these findings. We used the Mann Whitney-U Test to compare medians across the two groups (Ecolectivos investigators compared with external evaluators).” We rewrote the section to try to make it more clear about why we chose the Mann-Whitney test. We did not estimate an effect size in this cross-sectional study because we are not conducting hypothesis testing or estimating a treatment effect. We did provide confidence intervals since we know that our small sample does not represent a given population and are not attempting to estimate a measure of central tendency and our confidence around that measure if the survey were to be repeated. 

h. A detailed section for the qualitative part it essential

Reviewer response: We revised the qualitative section and provided references to describe the thematic analysis of the short responses that were provided in our online survey. See lines 202-208

i. Explain the modifications to the study

Reviewer response: The modifications are the results of an NIH-funded supplement that we received to add environmental health promotors and a dissemination component to the control group at the end of the study. Because this was not evaluated in the protocol for this study, and would be too confusing to explain, we removed it. 

6) There seems to be a misunderstanding, as the Results section includes aspects that should belong in the Methods section. 

Reviewer response: We have moved the two questions that were asked from the results section to the methods section (Lines 163-166). We have removed the scoring criteria descriptions; they are now in the methods section. We believe that is what the reviewer is referring to when stating that methods are in the results section. 

a. More detail on eligibility, analyses, and scoring methods are relevant.

Reviewer response: We have modified the methods section to address these concerns. 

b. How the data were collected or measured, details of survey questions, and the scales used for evaluation could be provided in Methods.

Reviewer response: We have modified the methods section to address these concerns. 

c. Correlations are also advisable - as well as potential regression analyses.

Reviewer response: Each of the domains are independent so we do not feel that correlations would be warranted, and these have not been reported in the literature with PRECIS-2 in other publications as far as we can determine. With only 25 participants, we feel that we do not have the statistical power to do a regression analysis, nor do we have a single outcome, instead we have 9 outcomes, thus we are underpowered to uses regression techniques. 

7) The formative phase baseline recruitment, involving door-to-door approaches, was considered more explanatory due to the additional effort required, while recruitment for the main trial was viewed as pragmatic, with village-level randomisation and community-based approaches that aligned with usual community engagement. Please elaborate.

Reviewer response: We have re-written the section to clarify the statement. 

8) The authors should consider checking their data again. For example, in Tables:

a. In Table 2, there are no descriptions of additional tests (e.g. chi-squared). Also check of how these influence the other variables.

Reviewer response: Given the small sample size of this exploratory study, we did not calculate X2 tests between Ecolectivos members and external evaluators since the cell sizes were even smaller (with some cells being = 0) than what is presented in Table 2 which presents an overall view of the 25 participants. We did not assess whether age, gender, race, ethnicity influenced how external evaluators versus Ecolectivos members selected scores in each of the 9 domains based on the rational describe in (6c) above.

b. In Table 3, please expand including effects, CIs and check outliers.

Reviewer response: Given the small sample size of this exploratory study, we did not conduct regression analyses to check the effects of demographic variables. Since we did not conduct inferential statistical analyses, we did not calculate CIs. We did not estimate an effect size in this cross-sectional study because we are not assessing a treatment effect. 

c. Considering the small sample, the IQRs can be double-checked

Reviewer response: Two investigators re-ran the data and we found two numbers that were rounded to whole numbers, representing the Likert scale, which we corrected. We infer that the reviewer is puzzled by the 25th or 75th percentile being the same as the 50th percentile, but that is indeed the case in which several of these domains-- if everyone ranked similarly, the 50th and the 25th or the 75th might be the same. 

d. In

---

## [Decision Letter · Decision Letter 1]

8 Dec 2024

PRECIS-2 used as an implementation science tool for global environmental health:

A cross-sectional evaluation of the Ecolectivos study protocol to reduce burning of household plastic waste in rural Guatemala

PONE-D-24-25892R1

Dear Dr. Thompson,

We’re pleased to inform you that your manuscript has been judged scientifically suitable for publication and will be formally accepted for publication once it meets all outstanding technical requirements.

Kind regards,

Thiago P. Fernandes, PhD

Academic Editor

PLOS ONE

Additional Editor Comments (optional):

Thank you for your valuable submission.

Upon reassessment, the authors have addressed all remaining concerns.

The only notable issue is the overly lengthy presentation of the Abstract, which could be simplified to improve its flow and clarity for readers.

I strongly encourage the authors to check and correct this during typesetting, as the Abstract is often the only section readers engage with. A concise yet robust approach would be more effective than extending its length unnecessarily.

Wishing you success with your study.

Reviewers' comments:

Reviewer's Responses to Questions

**Comments to the Author**

1. If the authors have adequately addressed your comments raised in a previous round of review and you feel that this manuscript is now acceptable for publication, you may indicate that here to bypass the “Comments to the Author” section, enter your conflict of interest statement in the “Confidential to Editor” section, and submit your "Accept" recommendation.

Reviewer #1: All comments have been addressed

2. Is the manuscript technically sound, and do the data support the conclusions?

Reviewer #1: Yes

3. Has the statistical analysis been performed appropriately and rigorously? 

Reviewer #1: Yes

4. Have the authors made all data underlying the findings in their manuscript fully available?

Reviewer #1: Yes

5. Is the manuscript presented in an intelligible fashion and written in standard English?

Reviewer #1: Yes

6. Review Comments to the Author

Reviewer #1: (No Response)

7. PLOS authors have the option to publish the peer review history of their article (what does this mean?). If published, this will include your full peer review and any attached files.

Reviewer #1: No

---

## [Editor Report · Acceptance letter]

16 Dec 2024

PONE-D-24-25892R1 

PLOS ONE

Dear Dr. Thompson, 

I'm pleased to inform you that your manuscript has been deemed suitable for publication in PLOS ONE. Congratulations! Your manuscript is now being handed over to our production team.

Kind regards, 

on behalf of

Dr. Thiago P. Fernandes 

Academic Editor

PLOS ONE